# Human health risk assessment of edible body parts of chicken through heavy metals and trace elements quantitative analysis

**Easmin Hossain[1], Meherun Nesha[2], Muhammed Alamgir Zaman Chowdhury[2], Syed Hafizur Rahman** ![ORCID][1]*

**1** Department of Environmental Sciences, Jahangirnagar University, Savar, Dhaka, Bangladesh,
**2** Agrochemical and Environmental Research Division, Institute of Food and Radiation Biology, Bangladesh Atomic Energy Commission, AERE, Ganakbari, Savar, Dhaka, Bangladesh

* hafizsr@juniv.edu

**Data Availability Statement:** All relevant data are within the manuscript and its Supporting information files.

## Abstract

Food safety and security have now been regarded as a significant emerging area within the food supply chain leading to scientific and public health concerns in the global world. The poultry sector is a substantial threat to heavy metal intoxication for Bangladeshi people due to contaminated drinking water and feed sources, as well as the poultry sector's surrounding environment and soil. This study was carried out to ascertain the residual concentrations of heavy metals (Pb, Cd) and trace elements (Cr, Fe, Cu, and Zn) in various edible chicken body parts (breast, liver, gizzard, heart, kidney, and brain) to observe the quality of the consumed chickens and to assess public health risk. Atomic absorption spectrometry (AAS) was used to check the content of toxic heavy metals and trace elements in 108 samples of 18 broiler chickens collected from six different locations of Dhaka North City Corporation markets in Bangladesh. The measured concentrations (mg/kg fresh weight) ranged from 0.33±0.2 to 4.6±0.4, 0.004±0.0 to 0.125±0.2, 0.006±0.0 to 0.94±0.4, 4.05±4.2 to 92.31 ±48.8, 0.67±0.006 to 4.15±2.7, and 4.45±0.62 to 23.75±4.3, for Pb, Cd, Cr, Fe, Cu, and Zn respectively. Except for Pb and Cu most of the investigated heavy metals and trace element levels in chickens were lower than the maximum allowable concentration (MAC) set by FAO/WHO and other regulatory agencies., The estimated level of Pb was nearly six times higher in the chicken brain. The estimated daily intake (EDI) values for all the studied metals were below the preliminary tolerated daily intake (PTDI). The target hazard quotient (THQ) values of the broiler chicken meat samples varied for adults and children, and the range was found to be 0.037–0.073 for Pb, 0.007–0.01 for Cd, 0.0–0.08 for Cr, 0.002–0.004 for Fe, for 0.00–0.002 Cu, and 0.004–0.008 for Zn, not exceeding the maximum level of 1 according to USEPA. The calculated THQ and total target hazard quotient (TTHQ) values were measured at less than one, suggesting that the consumption of chicken meat has no carcinogenic danger to its consumers. The Target carcinogenic risks (TCRs) of Pb, Cd, Cr, and Cu were within acceptable limits. The TCR values for children were, to some extent, higher than that of adults, which proposes that regular monitoring of both harmful and essential elements in chicken samples is necessary to determine whether or not any possible health risk

**Funding:** The author(s) received no specific funding for this work.

**Competing interests:** The authors have declared that no competing interests exist.

to consumers exists. In terms of health, this study demonstrated that consumers are chronically exposed to elemental contamination with carcinogenic and non-carcinogenic effects.

## Introduction

Food safety has become a severe global human health concern due to toxic heavy metal poisoning in the food chain [1]. Heavy metals have a specific weight greater than 5 g/cm$^3$ and are naturally present in the Earth's crust [2]. They are significant environmental contaminants that are harmful even in low concentrations. Heavy metals naturally exist in the soil, but concentrations beyond the acceptable limit are considered sources of environmental contamination [3]. Anthropogenic activities such as mining, exploitation of metal ore, industrialization, wastewater irrigation, agricultural activities, transportation, fuel combustion, iron and steel production, waste incineration, non-ferrous manufacturing, and cement kilns are releasing heavy metals in increasing amounts into the environment [4]. Heavy metals linger in the environment for a long time because they are difficult to break down [5]. They accumulate in living organisms due to dietary and non-dietary exposure if consumed over a long period. The acceptance of heavy metals by living beings is influenced by the bioavailability of heavy metals, nutritional variables, and the age of the organisms [6]. Some heavy metals are necessary for life, but some others, such as Lead (Pb), Chromium (Cr), and Cadmium (Cd) have several detrimental health effects. Very little of this HM is necessary for the body to function normally, but high concentrations might damage important organs like the kidneys [7]. Lead, for example, can affect a child's ability to learn and function cognitively, elevate blood pressure and cause coronary heart disease in adults [8]. Additionally, extreme Cd exposure will result in health problems such as skeletal difficulties [7]. As part of the sequelae leading to death rates, Chromium exposure can adversely affect the respiratory system, heart, gastrointestinal tract, haematology, liver, kidneys, and nervous system [9]. Because of the cumulative effects of dietary chicken, children are more at risk for Cd, Pb, and As than adults [10]. Trace elements are significant because of their necessity and toxic effects. When metal intake is exceptionally high, the essential elements might generate harmful consequences [11, 12]. According to World Health Organization research, long-term exposure to environmental contaminants causes 25% of human diseases [13]. Heavy metals present in minute amounts in the environment are biomagnified and form part of different food chains, where concentration elevates to levels that are poisonous to humans and other living things [14]. Usually, poultry meat and other edible body parts have several nutritional advantages as these items contain essential vitamins, protein, and minerals that enable people to maintain a healthy diet. Compared to red meat animals, chickens have a shorter supply chain and are superior feed efficiency converters [15]. For non-vegetable protein, people in densely populated nations like Bangladesh rely primarily on poultry meat and eggs. To fulfil the rising need for animal protein, Bangladesh has developed commercial poultry farming, which is considered a way to rapidly expand the production of this meat [1]. Poultry meat alone accounts for 37% of Bangladesh's overall meat output and (22–27) % of the nation's overall supply of animal protein [16]. Heavy metal contamination of poultry products may occur through feeds, drinking water, and chicken processing, a major problem [17]. Poultry farm owners and suppliers have endeavoured to meet the increased demand for chicken meat by keeping a consistent feed supply. In recent years, many chicken feed manufacturers in developing countries have been using tannery solid wastes, which are later converted to protein concentrates and used for poultry feed production

without adequate treatment, which contains a significant quantity of heavy metals [18]. Accumulating the metals in the chicken body parts may cause a potential risk to human health from consuming those contaminated chickens [19].

Furthermore, long-term intake of heavy metal-contaminated chickens may result in a toxic metal deposition in several vital organs, posing significant health hazards [20]. Most of the previous studies from Bangladesh investigated heavy metals from chicken meat as a whole (different chicken body parts were not considered) and found human health risks significantly. This study aimed to examine the levels of heavy metals such as Lead (Pb) and Cadmium (Cd) and trace elements, including Iron (Fe), Chromium (Cr), Copper (Cu), and Zinc (Zn) in six chicken body parts (muscle, liver, gizzard, heart, kidney, and brain) that are generally consumed by the Bangladeshi population and a risk assessment of both carcinogenic and non-carcinogenic human health concerns related to those chickens' ingestion.

## Materials and methods

### Study area

The research area was Dhaka North City Corporation (DNCC) of Dhaka district in Bangladesh. The area is 196.22 Sq. Km, located between 23°044′ and 23°054′ north latitudes and between 90°020′ and 90°028′ east longitudes (Fig 1). It is bounded by Dhamrai and Tongi on the north, Dhaka South City Corporation (DSCC) on the south, Narayanganj on the east, and Savar and Manikganj on the west. The number of populations in the studied area is about 6.10 million, and the average population density is about $31,115 Km^{-2}$. In Bangladesh, the overall farm, including farms of broilers and layers, is estimated to be over 114763 [21]. Around 56% of them are concentrated in Dhaka [22]. These farms provide chickens to the DNCC kitchen markets.

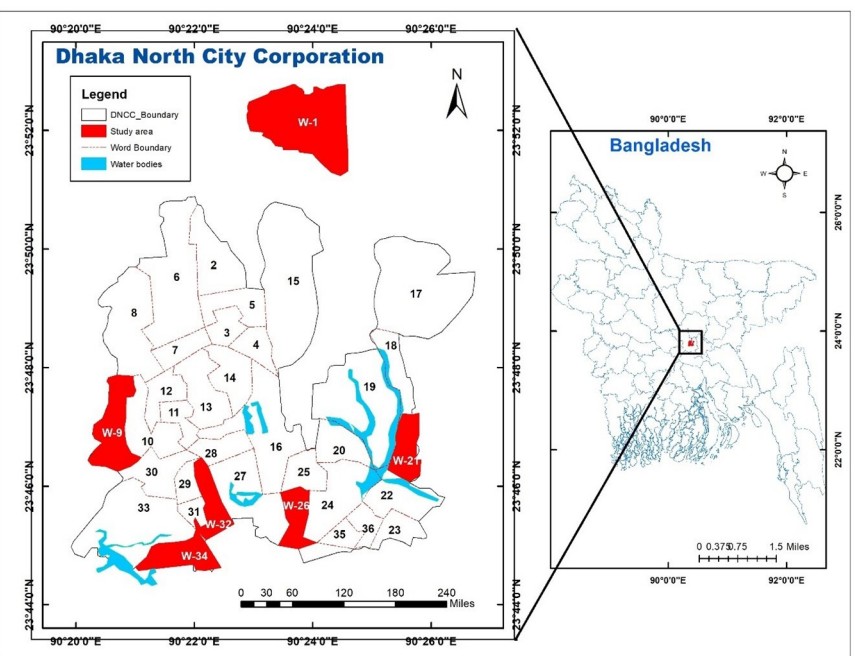

**Fig 1. Sampling location map of the study in Dhaka North City Corporation (DNCC).**

## Reagents and chemicals

Reference standards for Cu, Fe, Cd, Cr, Pb, and Zn were obtained from inorganic ventures (USA). $HNO_3$ and $HClO_4$, analytical-grade digestion chemicals utilized in this work, were bought from Merck (Supelco, Inc chemical industry company). The reagents were used without further purification.

## Sample collection and preservation

A multistage cluster sampling strategy was employed in this research where six different chicken markets were chosen from six wards of fifty-four DNCC wards, and three broiler chickens from each market (Uttara, Gabtoli, Madhya Badda, Karwan Bazar, Mohammadpur Town Hall Bazar, and Rayer Bazar markets) were selected. One hundred eight samples of six edible body parts (breast muscles, brain, liver, heart, kidney, and gizzard) of each chicken (6 markets×3 chickens from each market×6 edible body parts) were collected and stored at -20°C in plastic bags and brought to the laboratory of the Agrochemical and Environmental Research Division, Institute of Food and Radiation Biology, Atomic Energy Research Establishment (AERE), Bangladesh Atomic Energy Commission, Ganakbari, Savar, Dhaka for analysis. Neither human subject nor animal subject was employed in this research. The National Research Ethics Committee (NREC) of Bangladesh only provides approval/ certification for human/ animal subject-related research. As no living human/ animal body was used in the laboratory for analysis, ethical approval is not required for this work. Similar ethical approval exempt researches are also cited in the references (e.g., [2, 7]).

## Preparation and treatment of samples

The 108 samples were washed with distilled water to eliminate any contaminating particles. After that, the samples were cut into little chunks using a clean stainless-steel knife. Samples were dried in a continuous oven (OP100, LTE Scientific Ltd, Greenfield, Oldham, Great Britain) at 80ºC for 48 hours to achieve consistent weight. Once the drying was completed, the samples were ground into a fine powder with a ceramic mortar pestle, stored in polyethene bags, tagged, and kept cold and dry until acid digestion. Based on the samples' fresh and dry weights, the moisture contents of the samples were estimated.

## Acid digestion of samples

Three powdered samples (1 g each) of each sample were accurately weighed and put into porcelain crucibles to create three sample duplicates. A mixture of 10 ml nitric acid (65%) and perchloric acid (70%) (V: V = 3:1) was added to the powder samples and then placed on a hot plate (PHOENIX Instrument) at 160°C till the transparent solution was achieved [22]. Once the digestion process was completed, the crucibles were allowed to cool to room temperature. The contents were filtered using Whatman filter paper no. 42 after the samples were cooled [23]. Finally, deionized water was used to dilute each sample solution to a final volume of 50 ml. All glassware and containers were soaked in a 10% nitric acid solution for 24 hours before being repeatedly washed with tap water and then distilled water. A reagent blank solution was also prepared to avoid reagent contamination, where all the reagents except the sample were used to make the solution for each sample group.

## Preparation of working standards

Standard solutions with six different concentrations for each chosen metal were prepared to calibrate the instrument. Reference standard metals of Cu, Pb, Zn, Cr, Fe and Cd were

obtained from inorganic ventures in the USA. 50 ml of 0.0, 0.1, 0.2, 0.5, 1.0, and 2.0 mg/l of working standards of metals (Cu, Pb, Zn, Cr, Fe) and 0.0, 0.1, 0.2, 0.3, 0.4, 0.5 mg/l of working standards of Cd were made from their stock solution of 1000 mg/l. Deionized water was used to prepare each working standard.

## Instrument measurements

A flame atomic absorption spectrophotometer (model AA-7000, Shimadzu Corporation, Japan) was employed to investigate the extracts of the samples for the metals under investigation. In this spectrophotometer, background correction was done by a deuterium-arc lamp. The digested sample, in an aliquot, was injected into the air acetylene flame for Pb, Cd, Zn, Cr, Cu, and Fe using a Shimadzu autosampler ASC-7000. After every ten samples, a blank solution was run and analyzed by AAS before the studied samples, and the blank reading was subtracted from sample values to ensure that the equipment read only the exact values of our targeted metals.

The instrument configuration and operational conditions were followed according to the manufacturer's instructions. Table 1 summarizes the operating requirements for AAS.

In this investigation, metal concentrations (in ppm (mg/kg), fresh weight) were determined using Eq 1 as follows:

$$\text{Element, (ppm, mg/kg)} = R \times D/W \tag{1}$$

Where,

R = reading of the trace metal concentration in ppm from the digital scale of the instrument
D = Dilution factor of the prepared sample
W = weight of the sample

## Recovery analysis

Additionally, by spiking previously analyzed samples with varying quantities of aliquots of metal standards and reanalyzing the samples, a recovery analysis of the complete analytical method was conducted for metals in specified muscle samples. There were three replications of each determination. A recognized standard and a blank were run after every ten samples to assess the instrument's reliability. Acceptable metal recoveries for Pb, Cd, Cr, Fe, Cu, and Zn were 115.0%, 80.36%, 108.54%, 76.67%, 85.24% and 92.23%, respectively. The unit of measurement for heavy metal concentrations in mg/kg dry weight (mg/kg d wt.) is expressed on a dry tissue basis. The estimated moisture contents of the muscle and liver samples were 72.78± 2.5% and 69.74± 8.9%, respectively.

**Table 1. The wavelength, slit length, flame type, flow rate, and burner height of each metal.**

| Elements | Wavelength (nm) | Lamp Current (mA) | Slit (nm) | Flow (L/min) | Burner Height (mm) |
|:---:|:---:|:---:|:---:|:---:|:---:|
| Pb | 283.3 | 10 | 0.7 | 2.0 | 7 |
| Cd | 228.8 | 8 | 0.7 | 1.8 | 7 |
| Cr | 357.9 | 10 | 0.7 | 2.8 | 9 |
| Fe | 248.3 | 12 | 0.2 | 2.2 | 9 |
| Cu | 324.3 | 6 | 0.7 | 1.8 | 7 |
| Zn | 213.9 | 8 | 0.7 | 2.0 | 7 |

## Statistical analysis

Data were provided as mean ± Standard Deviation (SD) for each sample, and each analysis was performed in triplicate. The statistical analyses were conducted using the statistical package SPSS (IBM SPSS Statistics version: 28.0.1.0 (142)) and Microsoft Excel 2019.

## Human health risk assessment

The process of determining the probability of adverse human health consequences on exposure to specific chemical agents for a certain period is known as human-health risk assessment. Humans can come into contact with contaminants through a variety of routes. These routes may be cutaneous, respiratory, or gastrointestinal in nature. This study evaluated the risks of ingesting heavy metals from the consumption of edible chicken meat [24].

**Estimated Daily Intakes (EDI) of metals.** The EDI of particular heavy metals were calculated using the mean metal content of samples, a daily chicken intake rate, and an individual's body weight. Eq (2) was used to calculate according to [25].

$$EDI = (DFC \times MC)/BW \qquad (2)$$

DFC is the daily food consumption rate (g/person/day). In Bangladesh, for adults and children, the DFC are 17.4 and 8.3 g/person/day, respectively. This number was collected from the "Report of the Household Income and Expenditure Survey 2010," published in 2010 [26]. MC denotes the mean metal content in chicken samples (mg/kg fresh weight). BW stands for body weight (60 kg for adults and 27 kg for children) [10].

**Target Hazard Quotient (THQ).** Target hazard quotients (THQs) were used in this investigation to analyze the non-carcinogenic health risks related to meat intake, and calculations were performed based on the assumption for an integrated USEPA risk analysis, as shown in Eq 3 taken from [27].

$$THQ = \frac{EFr \times ED \times FIR \times MC}{RfD \times BW \times AT} \times 10^{-3} \qquad (3)$$

Where THQ is the dimensionless target hazard quotient, EFr is the exposure frequency (365 days per year), and ED is the exposure duration (70 years for adults and 14 years for children) comparable to the typical human lifetime [10]. FIR stands for food ingestion rate (g/person/day), MC stands for element concentration in samples (mg/kg fresh weight), BW stands for average body weight (adult, 60 kg; children, 27 kg), AT stands for averaging time for non-carcinogens (365 days/year number of exposure years), and RfD stands for oral reference dose (mg/kg/day) [10]. The RfDs for Pb, Cd, Cr, Fe, Cu and Zn are 0.0035, 0.001, 0.003, 0.7, 0.04, and 0.3 based on mg/kg BW/day, respectively [28]. The RfDs calculate the daily exposure to which people may be exposed continuously throughout a lifetime without a real risk of adverse consequences. If the THQ is equal to or more than one, there is a possible health concern [29], and appropriate treatments and preventative measures should be implemented. Exposure to two or more hazardous components has been linked to addictive and interaction effects [30]. The sum of the THQ values for each metal was used to calculate the total THQ (TTHQ, individual meat item) of heavy metals for each type of meat as follows [31]:

$$TTHQ = THQ\ metal\ 1 + THQ\ metal\ 2 + THQ\ metal\ 3 + \cdots + THQ\ metal\ n \qquad (4)$$

The USEPA established the Hazard Index (HI) technique to evaluate the possible risk for non-carcinogenic consequences provided by many elements [27]. HI was calculated for a

particular receptor/pathway combination (e.g., diet) as follows:

$$HI = TTHQ1 + TTHQ2 + TTHQ3 + \cdots + TTHQn \tag{5}$$

When HI is more than one, there is a danger of health problems.

**Target Carcinogenic Risk (TCR).**  Carcinogen risks were estimated as the cumulative chance that a person would get cancer if exposed to that potential carcinogen (i.e., incremental or excess individual lifetime cancer risk) over their lifespan [27]. Carcinogens have allowable levels of risk ranging from $10^{-4}$ to $10^{-6}$. The following equation is used to calculate the target cancer risk (lifetime cancer risk) [27]:

$$TCR = \frac{EFr \times ED \times R \times MC \times CSFo}{BW \times AT} \times 10^{-3} \tag{6}$$

Where, TCR is the target cancer risk or the lifetime risk of cancer, and CSFo is the oral carcinogenic slope factor, which was 0.0085 $(mg/kg/day)^{-1}$ for Pb, 0.38 $(mg/kg/day)^{-1}$ for Cd, 0.5 $(mg/kg/day)^{-1}$ for Cr, and 1.5 $(mg/kg/day)^{-1}$ for Cu from the Integrated Risk Information System database [28, 32].

## Results and discussion

### Concentrations of heavy metals in different body parts of the chicken

Pb, Cd, Cr, Fe, Cu, and Zn (mg/kg fresh weight) were determined in the muscle, liver, gizzard, heart, kidney, and brain of chickens collected from DNCC presented in Table 2. On a fresh weight basis, all metal concentrations were estimated. In broiler chicken samples, the concentrations of several heavy metals and trace elements ranged from 0.33±0.2 to 4.6±0.4 mg kg$^{-1}$, 0.004±0.0 to 0.125±0.2 mg kg$^{-1}$, 0.006±0.0 to 0.94±0.4 mg kg$^{-1}$, 4.05±4.2 to 92.31±48.8 mg kg$^{-1}$, 0.67±0.006 to 4.15±2.7 mg kg$^{-1}$, and 4.45±0.62 to 23.75±4.3 mg kg$^{-1}$ Pb, Cd, Cr, Fe, Cu, and Zn respectively. The maximum permissible levels (mg/kg) for chicken meat and offal by international regulatory agencies are shown in Table 3. The data on heavy metals and trace elements concentration in this investigation were compared to data from previous research conducted in Bangladesh and other regions of the world that have been published, and the results are provided in Table 4.

**Lead (Pb).**  Pb is a toxic heavy metal that causes significant environmental damage and health issues. The human body is affected by lead in a variety of ways. Acute lead poisoning causes appetite loss, headaches, high blood pressure, nausea, kidney failure, lethargy, insomnia, arthritis, schizophrenia, and dizziness. Long-lasting lead exposure has been linked to intellectual disabilities, birth deformities, schizophrenia, autism, allergies, dyslexia, weight loss, hyperactivity, disability, muscle weakness, neurological damage, renal damage, and even demise [33]. Lead is a non-essential element associated with neurotoxicity, nephrotoxicity, and various other health issues [34]. The mean±SD Pb values in the studied muscle, liver, gizzard, heart, kidney and brain samples were 0.657±0.17, 0.494±0.3, 0.747±0.2, 0.695±04, 0.998±0.6, and 3.454±1.2 (in mg/kg FW) are shown in (Fig 2). The lead concentration levels in different body parts of the tested chickens followed the order; brain > kidney > gizzard > heart > muscle> liver. According to the current investigation results, lead (Pb) concentration in the brain was much greater than in the other organs. The maximum Pb concentration of 4.943 mg/kg was found in brain samples collected from the Madhya Badda Kacha Bazar area, and the lowest value was obtained from liver samples collected from the Mohammadpur area of DNCC. The concentration of lead in chicken muscle samples was higher compared to the studies done by [17, 35–38] and lower compared to that of [15, 39–41]. Other investigators recorded the same results (0.49 mg kg-1) from liver samples [12]. The Pb content in chicken gizzards from the

**Table 2. The concentration of different heavy metals (mg/kg fresh weight) in broiler chicken from other regions of Dhaka North City Corporation (mean ± SD range).**

| Sample types | Uttara | Gabtoli | Badda | Karwan Bazar | Mohammadpur | Rayer Bazar | Min~ Max | Overall mean |
|---|---|---|---|---|---|---|---|---|
| **Lead (Pb)** | | | | | | | | |
| **Muscle** | 0.445±0.02 | 0.734±0.1 | 0.881±0.09 | 0.679±0.1 | 0.656±0.2 | 0.547±0.1 | 0.428–0.963 | 0.657±0.2 |
| **Liver** | 0.391±0.1 | 0.552±0.1 | 0.772±0.2 | 0.486±0.3 | BDL | 0.762±0.7 | 0.0–1.178 | 0.494±0.4 |
| **Gizzard** | 0.555±0.1 | 0.737±0.1 | 0.881±0.3 | 0.655±0.2 | 0.645±0.1 | 1.012±0.2 | 0.472–1.230 | 0.747±0.2 |
| **Heart** | 0.506±0.3 | 0.328±0.2 | 1.007±0.4 | 0.689±0.2 | 0.537±0.0 | 1.104±0.3 | 0.126–1.407 | 0.695±0.4 |
| **Kidney** | 1.253±0.3 | 0.650±0.2 | 0.906±0.5 | 1.607±1.3 | 0.726±0.1 | 0.844±0.1 | 0.492–3.134 | 0.998±0.6 |
| **Brain** | 3.903±0.3 | 3.510±0.7 | 4.943±0.99 | 3.964±1.1 | 2.019±0.5 | 2.382±0.9 | 1.487–13.851 | 3.454±1.2 |
| **Cadmium (Cd)** | | | | | | | | |
| **Muscle** | 0.035±0.0 | BDL | BDL | 0.034±0.0 | 0.023±0.0 | 0.037±0.0 | 0.004–0.078 | 0.021±0.02 |
| **Liver** | 0.058±0.1 | 0.069±0.0 | 0.078±0.0 | 0.102±0.0 | 0.107±0.0 | 0.075±0.0 | 0.038–0.134 | 0.082±0.02 |
| **Gizzard** | BDL | BDL | 0.007±0.0 | 0.049±0.0 | 0.047±0.0 | 0.084±0.0 | 0.007–0.116 | 0.055±0.04 |
| **Heart** | BDL | BDL | BDL | 0.009±0.0 | 0.002±0.0 | BDL | 0.002–0.009 | 0.004±0.002 |
| **Kidney** | 0.016±0.0 | 0.004±0.0 | 0.032±0.0 | 0.006±0.0 | 0.012±0.0 | 0.017±0.0 | 0.002–0.045 | 0.015±0.01 |
| **Brain** | BDL | 0.037±0.0 | 0.041±0.0 | BDL | BDL | BDL | 0.037–0.044 | 0.04±0.004 |
| **Chromium (Cr)** | | | | | | | | |
| **Muscle** | BDL | BDL | BDL | 0.281±0.02 | 0.826±0.0 | BDL | 0.253–0.826 | 0.236±0.3 |
| **Liver** | 0.550±0.0 | 0.920±0.0 | BDL | 0.473±0.0 | BDL | BDL | 0.473–0.920 | 0.648±0.3 |
| **Gizzard** | 0.223±0.0 | BDL | BDL | BDL | BDL | BDL | 0.223–0.223 | 0.223±0.05 |
| **Heart** | BDL | BDL | BDL | 0.286±0.0 | 0.006±0.0 | 0.364±0.0 | 0.006–0.364 | 0.219±0.2 |
| **Kidney** | BDL | 0.147±0.0 | BDL | 0.053±0.0 | BDL | BDL | 0.053–0.147 | 0.1±0.06 |
| **Brain** | 0.569±0.0 | 0.944±0.4 | 1.006±0.3 | 0.777±0.0 | BDL | 0.569±0.0 | 0.569–1.412 | 0.899±0.3 |
| **Iron (Fe)** | | | | | | | | |
| **Muscle** | 4.682±1.2 | 5.154±5.0 | 2.701±3.8 | 5.576±2.6 | 10.277±1.6 | 7.058±1.2 | 1.111–11.934 | 6.256±3.4 |
| **Liver** | 76.879±23.8 | 56.254±16.4 | 52.936±14.6 | 39.967±6.4 | 92.319±48.8 | 77.172±12.2 | 32.54–142.825 | 65.921±27.5 |
| **Gizzard** | 20.524±5.5 | 19.152±5.6 | 17.281±4.8 | 22.663±3.3 | 18.262±8.3 | 37.107±18.6 | 10.304–58.599 | 22.498±10.4 |
| **Heart** | 33.449±12.1 | 14.727±7.4 | 28.818±5.7 | 28.502±0.9 | 26.364±1.1 | 29.887±0.8 | 7.645–46.333 | 26.958±8.0 |
| **Kidney** | 43.947±13.2 | 53.432±28.4 | 32.727±6.9 | 45.029±15.4 | 44.656±8.2 | 42.518±5.6 | 24.747–82.962 | 43.718±14.1 |
| **Brain** | 79.619±57.0 | 25.490±5.5 | 26.996±1.6 | 27.601±8.5 | 41.239±37.9 | 25.526±12.8 | 17.671–135.876 | 37.745±31.4 |
| **Copper (Cu)** | | | | | | | | |
| **Muscle** | 0.141±0.03 | 0.158±0.0 | 0.068±0.01 | 0.174±0.1 | 0.325±0.1 | 0.228±0.0 | 0.018–0.411 | 0.191±0.1 |
| **Liver** | 2.380±0.4 | 2.278±0.3 | 2.980±1.3 | 2.373±0.6 | 1.926±0.3 | 2.368±0.3 | 1.59–4.448 | 2.384±0.6 |
| **Gizzard** | 0.557±0.2 | 0.501±0.1 | 0.429±0.1 | 0.470±0.2 | 0.315±0.2 | 0.543±0.2 | 0.152–0.752 | 0.469±0.2 |
| **Heart** | 2.785±0.5 | 1.281±0.4 | 2.409±0.7 | 2.696±0.3 | 2.631±0.2 | 2.871±0.4 | 0.832–7.264 | 2.445±0.7 |
| **Kidney** | 2.303±0.6 | 1.421±0.6 | 1.903±0.2 | 2.440±0.4 | 1.931±0.5 | 2.134±0.1 | 0.859–2.978 | 2.007±0.5 |
| **Brain** | 3.454±2.7 | 2.930±0.3 | 3.112±0.7 | 2.793±0.9 | 2.387±0.5 | 2.579±0.6 | 1.564–6.592 | 2.876±1.1 |
| **Zinc (Zn)** | | | | | | | | |
| **Muscle** | 4.452±0.6 | 4.530±0.6 | 5.576±0.8 | 7.953±6.8 | 6.078±0.9 | 6.311±1.2 | 4.452–7.953 | 5.817±2.7 |
| **Liver** | 16.045±2.6 | 18.301±1.5 | 23.545±11.1 | 16.986±0.6 | 17.453±3.5 | 23.755±4.3 | 16.045–23.755 | 19.347±5.4 |
| **Gizzard** | 16.599±2.9 | 18.010±0.7 | 16.200±2.8 | 16.455±2.1 | 11.786±5.3 | 20.217±1.6 | 11.786–20.217 | 16.545±3.6 |
| **Heart** | 16.625±1.2 | 7.737±2.6 | 16.601±3.3 | 16.279±0.6 | 15.508±0.2 | 17.155±2.2 | 0.958–19.610 | 14.984±3.8 |
| **Kidney** | 16.131±5.3 | 10.245±4.2 | 14.063±2.6 | 16.432±2.5 | 15.524±2.9 | 15.095±0.6 | 6.277–21.963 | 14.582±3.6 |
| **Brain** | 19.491±11.6 | 13.107±1.3 | 15.054±1.7 | 13.117±0.6 | 12.466±1.9 | 10.492±0.95 | 9.869–32.814 | 13.954±5.0 |

BDL means below the detection limit

**Table 3. Maximum permissible levels (mg/kg) limits in chicken by worldwide regulatory bodies.**

| Name of the Regulatory body | Country | Maximum Permissible level (mg/kg) limits | | | | | | References |
|---|---|---|---|---|---|---|---|---|
| | | Pb | Cd | Cr | Fe | Cu | Zn | |
| **FAO/WHO, 2002** | - | 0.1 | Meat-0.05 Liver-0.5 | 1.0 | 180 | 0.4 | 150 | [56] |
| **Codex Alimentarius, 2009** | United States | Meat-0.1 Offal-0.5 | - | - | - | - | - | [57] |
| **Australia New Zealand Food Authority, 2015** | Australia & New Zealand | Meat-0.1 Offal-0.5 | - | - | - | 200 | - | [58] |
| **European Communities, 2006** | European Countries | Meat-0.1 | Meat-0.05 Liver-0.5 | - | - | - | - | [59] |
| **Food Safety and Standards Authority of India, 2011** | India | Meat-0.1 Offal-0.5 | - | - | - | - | - | [60] |
| **Food Safety Authority of Ireland, 2009** | Ireland | Meat-0.1 Offal-0.5 | Meat-0.05 Liver -0.5 Kidney-1.0 | - | - | - | - | [61] |
| **CN, 2005** | China | 0.2 | Meat-0.1 Liver-0.5 | - | - | - | - | [62] |
| **JECFA, 2005** | - | 0.1 | 0.1 | - | - | 0.1 | - | [63] |
| **Export Inspection Council India, 2017** | India | Meat-0.1 Offal-0.5 | Meat-0.05 Liver-0.5 Kidney-1.0 | - | - | - | - | [64] |
| **The Egyptian Organization for Standardization and Quality Control** | Egypt | Meat-0.1 Offal-0.5 | Meat-0.05, Offal-1.0 | - | Meat- 15.0, Offal- 20.0 | Meat and offal- 15.0 | - | [65] |

Rayer Bazar region was higher than that in chicken gizzards collected from any other location, even it is less than (0.013.22 mg.kg-1) obtained by [42]. The MAC value of 0.1000 mg/kg concentration set by several regulatory agencies was found to be exceeded in all of the muscle samples that were collected (Table 3), and a similar trend was also seen in other samples (excluding liver), which exceeded the MAC value of 0.5 mg/kg. When comparing the Pb concentrations of the samples, statistically significant differences (p 0.05) were found. The significant bioaccumulation features of Pb in chicken brain tissues may account for the higher Pb levels in brain samples.

**Cadmium (Cd).** The toxic effects of cadmium are most detrimental to the liver and kidneys, accumulating in greater concentrations in the proximal tubular cells [43]. Cadmium may accumulate in the human body, resulting in renal, lung, hepatic, skeletal, and reproductive issues and cancer [6]. The mean ±SD Cadmium values in the muscle, liver, gizzard, heart, kidney and brain samples were 0.021±0.02, 0.082±0.02, 0.055±0.04, 0.004± 0.003, 0.015± 0.01, 0.04±0.004 mg/kg (Fig 2). Cadmium concentrations in different chicken body parts were as follows: liver > gizzard > brain > muscle > kidney> heart (Fig 2). The results of this study

**Table 4. Comparison of elemental concentrations (mg/kg FW) in chicken meat with the reported values in the literature.**

| Region | Pb | Cd | Cr | Fe | Cu | Zn | References |
|---|---|---|---|---|---|---|---|
| **Bangladesh** | 0.37 | 0.23 | 2.17 | N/A | 1.99 | N/A | [10] |
| **Bangladesh** | 0.75 | 0.01 | 0.223 | 16.87 | N/A | 10.37 | [32] |
| **Saudi Arab** | 2.72 | 0.46 | N/A | 66.33 | 6.56 | 10.37 | [68] |
| **Nigeria** | 0.215 | 0.016 | N/A | N/A | N/A | 1.57 | [72] |
| **Bangladesh** | 0.17 | 0.03 | 1.4 | N/A | 2.5 | N/A | [73] |
| **Egypt** | 0.25 | 0.03 | N/A | 6.77 | 0.15 | N/A | [74] |
| **Turkey** | 0.40 | 0.006 | 0.07 | 8.2 | 1.2 | 19.9 | [11] |
| **France** | 0.15 | N/A | 0.03 | N/A | 0.6 | 16.23 | [75] |
| **Pakistan** | 3.1 | 0.31 | N/A | N/A | 12.86 | 28.52 | [45] |
| **Bangladesh** | 0.657 | 0.021 | 0.236 | 6.13 | 0.182 | 5.817 | This study |

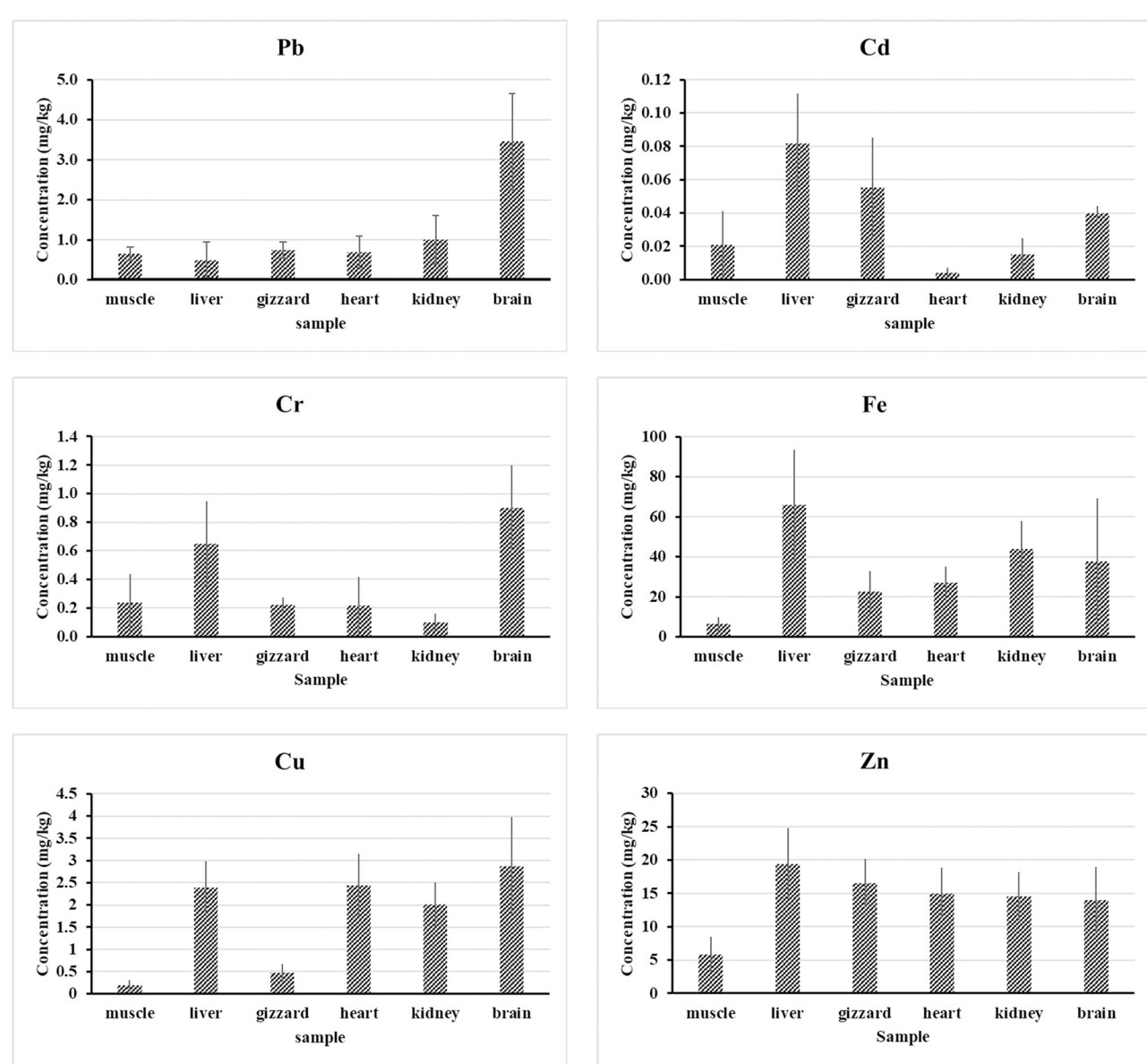

**Fig 2. Heavy metals (Pb, Cd), and trace elements (Cr, Fe, Cu, and Zn) concentrations in different body parts of the chicken.** Results are expressed as mean ± SD (average of 18 samples).

showed that Cd levels in the liver (0.058±0.1, 0.069±0.0, 0.078±0.0, 0.102±0.0,0.107±0.0, and 0.075±0.0 mg/kg respectively) were significantly higher than the chicken's other edible body parts collected from the different study area (Table 2 and Fig 2). Our findings were consistent with the study of [44], who observed higher concentrations of Cd in the liver samples than in the chicken muscles. According to [45], the concentrations were also higher in the liver than in the muscles. Cadmium concentrations in different organs (BDL– 0.134 mg/kg) in this study are higher than previous results ((BDL)– 0.037 mg/kg) in [46]; (0.019–0.061 mg/kg) in [43]; (0.019–0.099 mg/kg) in [47] and lower than (0.45–2.23 mg/kg) in [48], and (0.059–9.36) in [49]. In comparison to the other organs, the kidney samples showed a lower concentration (0.015 mg kg$^{-1}$) than other organs, which are lower than in the study [50]. The kidney samples indicated lower amounts as compared to the other organs like the muscle (0.021 mg kg$^{-1}$), liver

(0.082 mg kg$^{-1}$), gizzard (0.055 mg kg$^{-1}$) and brain (0.04 mg kg$^{-1}$). The overall mean concentration of Cd from the gizzard samples (0.055± 0.04 mg/kg) was higher than [35] (0.02± 0.01 mg/kg) but lower than [42], (0.01± 0.00 to 1.02± 0.14 mg/kg).

**Chromium (Cr).** Through the lungs and digestive tract, hexavalent chromium compounds absorb faster than trivalent chromium compounds. Increased chromium compounds can block erythrocyte glutathione reductase, reducing humans' ability to convert methemoglobin to haemoglobin [51]. The mean concentrations level for Chromium residues in muscle, liver, gizzard, heart, kidney, and brain samples were presented in (Fig 2). The mean ±SD values of Cr in the muscle, liver, gizzard, heart, kidney, and brain samples were 0.236±0.3, 0.648±0.3, 0.223±0.05, 0.219±0.2, 0.1±0.06, 0.899±0.3 (in mg/kg). The concentration of chromium in the chicken's different body parts was; brain > liver > muscle > gizzard > heart > kidney.

In the current investigation, the chromium content ranged from BDL to 1.412 mg/kg. Using tannery waste as feed-in poultry farming, a common practice in most of Bangladesh's agroecological zones with high Cr levels, may be responsible for the heightened Cr levels in the chicken meat samples [52]. The mean Chromium values in the muscle were lower than the results (2.17 ± 0.66 mg kg$^{-1}$) obtained by [10], (0.56 mg kg$^{-1}$) obtained by [40], (2.4±0.75 mg kg$^{-1}$) obtained by [37], (3.976±2.937 mg kg$^{-1}$) obtained by [53] and (3.6±1.3 mg kg$^{-1}$) obtained by [41]. The mean concentration of Chromium obtained in this study is higher than in the study of [36, 38, 54]. The highest level of Cr was detected in chicken brain samples (1.412 mg/kg), which is less than the finding obtained by [53]. Compared to results measured in the chicken gizzard from the study by [42] (0.38 and 2.33 mg.kg$^{-1}$) in Nigeria, the Cr content (0.223 mg kg$^{-1}$) in this study's chicken gizzard was lower. According to FAO/WHO, 2000 [55], 1.0 mg/kg of Cr is the maximum permissible limit in chicken meat. The concentration of Cr in the brain sample taken from the Badda region was significantly higher than the maximum allowable level (shown in Table 3).

**Iron (Fe).** The synthesis of RBC and haemoglobin requires Fe. Protein gets saturated when iron levels rise after consuming a large amount of food. Excess iron circulates throughout the bloodstream as free iron, which is poisonous to the organs it affects [66]. It is widely known that enough Fe consumption in the diet is crucial to reducing the prevalence of anaemia [67]. The mean ±SD values of Fe in the muscle, liver, gizzard, heart, kidney, and brain samples were 6.256±3.4, 65.921±27.5, 22.498±10.4, 26.958±8.02, 43.718±14.1, 37.745±31.4 (in mg/kg). The Iron concentrations in the various chicken body parts were; liver > kidney > brain > heart > gizzard > muscle. It shows that the liver and kidney samples contained a much higher level of Fe than other edible body parts of the chicken (Fig 2). Another researcher achieved the same results from chicken meat samples [29]. Post-examination of inebriated animals demonstrated gastrointestinal effects of more than 15.0 μg/g in meat and 20.0 μg/g in poultry offal [35]. The obtained mean value for chicken muscle (6.256 mg/kg) was greater than those of [54], but they were lower than the results obtained by [36, 40, 68]. All of the samples studied (100%) were found to be within the allowed limits and were pronounced safe for ingestion by humans. The obtained mean concentration of Fe in the liver sample (65.921mg/kg) is lower than the obtained result of [12, 36] and higher than the result of [50]. The findings indicated that, of all the chicken body parts, the liver had the greatest average and individual heavy metal concentrations, except Pb and Cr in a few areas, indicating that the liver has a far larger capability for bioaccumulation than the other parts (Table 2). However, in our experiment, all of the assessed heavy metals' average concentrations in the six chicken body parts that were examined followed the descending sequence of liver > kidney > brain > heart > gizzard > muscle, except Pb. The livers had the highest quantity of heavy metals, according to the prior study, when compared to other body organs [12, 17, 42].

**Copper (Cu).**   Copper is a crucial trace element that must be consumed in moderation for good health. Copper is required as a cofactor by various oxidative and reductive enzymes. The current Recommended Dietary Allowance (RDA) of Cu for adults in the United States and Canada is 9 mg/day, with a tolerated upper intake level (UL) of 10 mg/d for adults [69]. Although Copper is a necessary nutrient, this trace metal can damage the kidneys and liver if it is present in high doses [70]. The mean ±SD of Copper in the muscle, liver, gizzard, heart, kidney, and brain samples were 0.191±0.1, 2.384±0.6, 0.469±0.2, 2.445±0.7, 2.007±0.5, 2.876±1.1 (in mg/kg FW) are presented by (Fig 2). Copper concentrations in the various chicken body parts were; brain > heart > liver > kidney > gizzard > muscle.

The findings for the meat sample were more significant than the results of [17, 54] but lower than the results of [36, 40, 41, 48]. The Cu concentrations reported in tissues were higher than the recently published data obtained by [49] (2.34 mg/kg for the liver and 1.43 mg/kg for muscle), and when compared to FAO/WHO and EC standards, the results are significantly more. The result obtained by [29] (0.35± 0.003 mg/kg) was lower than the concentration of Cu that was obtained from this investigation (0.469± 0.2 mg/kg). The same observation was reported in the gizzard samples of slaughtered poultry by [42] in Agbor.

**Zinc (Zn).**   Zinc is a nutrient that promotes development. Its absence harms the growth of many animal species, including humans. Because zinc is required for protein and DNA synthesis and cell division, Zn's growth effect is thought to be linked to its impact on protein synthesis [71]. The recommended dietary allowance (RDA) for zinc for men is 11 mg, while it is 8 mg for women. The mean ±SD of Zinc values for the muscle, liver, gizzard, heart, kidney, and brain samples were 5.817±2.7, 19.347±5.4, 16.545±3.6, 14.984±5.0, 14.582±3.6, 13.954±5.0 (in mg/kg FW), respectively in (Table 2). The zinc contents in the various chicken body parts were; liver > gizzard >heart > kidney > brain > muscle. The mean concentrations of Zn in different study areas are shown in (Fig 2). The liver tissue had the highest zinc concentration (23.755±4.3 mg/kg), while the breast meat of chicken had the lowest concentration (4.452±0.6 mg/kg). In comparison to [38], the Zn concentrations of this study were low [45]. The Zn concentrations of this study were higher than the result from [54]. Zinc concentrations (6.277–21.963 mg/kg) in chicken kidneys from this investigation were lower than in the study [49]. The readings in the research samples were all below the acceptable limit (150 mg/kg) established by the EPA (ANZFA). In Table 4, the average Zn concentration (4.45±0.62 to 23.75 ±4.3) of this investigation was lower than that obtained (6.12–33.21 mg/kg) by [42] in Nigeria (27.93–36.93 mg/kg) by [68] in Saudi Arabia, and (26.67±0.25–28.67±0.34 mg/kg) by [71] in Bangladesh.

## Health risk assessment

**Estimated daily intake and maximum tolerable daily intake.**   The route and degree of exposure define a group's health risk or hazard. As a result, determining the degree of exposure through identifying pollutant routes to target groups is critical. The significant pathways of metal exposure to human health are ingestion, inhalation, and skin contact. The food chain or ingestion is the most significant of these channels. The current study looked at the Pb, Cd, Cr, Fe, Cu, and Zn ingestion route, which is thought to involve chicken consumption. The EDI values for certain metals were estimated using the average metal content in chicken and a person's specific consumption rate. Table 5 shows the EDI and preliminary tolerated daily intake (PTDI) values for the metals studied.

The EDI of the analyzed heavy metals through chicken meat consumption suggests that average consumption amounts of chicken meat are not a health concern because the resultant EDI value is less than the FAO/WHO recommended heavy metals intake values [10, 12].

**Table 5. Estimated dietary intakes (EDI) ($\mu g\ kg^{-1}\ BW^{-1}\ day^{-1}$) of heavy metals from the consumption of chicken meats by the people of Bangladesh.**

| Sample ID | Estimated dietary intake (EDI) | | | | | | | | | | | |
|---|---|---|---|---|---|---|---|---|---|---|---|---|
| | Pb | | Cd | | Cr | | Fe | | Cu | | Zn | |
| | Adult | Child | Adult | Child | Adult | Child | Adult | Child | Adult | Child | Adult | Child |
| Uttara | 0.129 | 0.137 | 0.010 | 0.011 | - | - | 1.358 | 1.439 | 0.041 | 0.043 | 1.291 | 1.369 |
| Gabtoli | 0.213 | 0.226 | - | - | - | - | 1.495 | 1.585 | 0.046 | 0.049 | 1.314 | 1.392 |
| Badda | 0.255 | 0.271 | - | - | - | - | 1.175 | 1.245 | 0.020 | 0.021 | 1.617 | 1.714 |
| Karwan Bazar | 0.197 | 0.209 | 0.010 | 0.010 | 0.081 | 0.086 | 1.617 | 1.714 | 0.050 | 0.053 | 2.306 | 2.445 |
| Mohammadpur | 0.190 | 0.202 | 0.007 | 0.007 | 0.239 | 0.254 | 2.980 | 3.159 | 0.094 | 0.100 | 1.763 | 1.869 |
| Rayer Bazar | 0.159 | 0.168 | 0.011 | 0.011 | - | - | 2.047 | 2.170 | 0.066 | 0.070 | 1.830 | 1.940 |
| Mean | 0.190 | 0.202 | 0.006 | 0.007 | 0.053 | 0.057 | 1.779 | 1.885 | 0.053 | 0.056 | 1.687 | 1.788 |
| PTDI | 3[c] | | 0.66[c] | | 2.8[a] | | 800[c] | | 166.7[b] | | 1000[c] | |

[a] RDA (1989) [76],

[b] DRI, (2001) [77],

[c] JECFA (2003) [78]

**Non-carcinogenic health risk.** THQ was used to evaluate non-cancerous health hazards associated with adult residents' consumption of hazardous metals-loaded chicken. The THQ is the ratio of the examined metal's dosage to the reference dosage level for a similar metal; if the ratio exceeds 1, population exposure is thought to have adverse health consequences that are not carcinogenic [69]. Table 6 shows the THQs of the six metals studied for the chicken meat samples for children and adults. In the meat sample of chickens, the THQ of most of the examined heavy metals was lower than 1, showing that exposure to a single metal through chicken food does not constitute a substantial health danger. For ingestion of the examined meat samples, TTHQ (sum of individual metal THQ) ranged from 0.054 to 0.153 for adults and from 0.058 to 0.163 for children, depending on the metals taken into account (Table 6). THQs values detected from the study areas indicated that the ingestion of chicken does not pose a cariogenic health risk for both children and adults. The THQ for the metals from the chicken meat samples consumption is in decreasing order of Pb > Cr > Cd > Zn > Fe > Cu and Pb > Cr > Zn> Cd > Fe > Cu for the children and adults, respectively.

**Target Carcinogenic Risk (TCR).** Target carcinogenic risks (TCRs) of Pb, Cd, Cr, and Cu were computed based on food consumption since these heavy metals can induce carcinogenic and non-carcinogenic risks depending on their exposure level. Pb, Cd, Cr, and Cu TRs attributable to chicken meat intake were estimated and presented in Fig 3. The mean TCR values of Pb, Cd, Cr, and Cu, attributable to chicken meat intake, were 9.7E-06, 1.4E-05, 1.6E-04,

**Table 6. Non-carcinogenic (THQ) health risks of heavy metals and trace elements due to consuming chicken meat from DNCC of Bangladesh.**

| Sample ID | Target hazard quotients (THQ) | | | | | | | | | | | | | |
|---|---|---|---|---|---|---|---|---|---|---|---|---|---|---|
| | Pb | | Cd | | Cr | | Fe | | Cu | | Zn | | Total TTHQ | |
| | Adult | Child | Adult | Child | Adult | Child | Adult | Child | Adult | Child | Adult | Child | Adult | Child |
| Uttara | 0.037 | 0.039 | 0.010 | 0.011 | 0.000 | 0.000 | 0.002 | 0.002 | 0.001 | 0.001 | 0.004 | 0.005 | 0.054 | 0.058 |
| Gabtoli | 0.061 | 0.064 | 0.000 | 0.000 | 0.000 | 0.000 | 0.002 | 0.002 | 0.001 | 0.001 | 0.004 | 0.005 | 0.068 | 0.073 |
| Badda | 0.073 | 0.077 | 0.000 | 0.000 | 0.000 | 0.000 | 0.002 | 0.002 | 0.000 | 0.001 | 0.005 | 0.006 | 0.081 | 0.085 |
| Karwan Bazar | 0.056 | 0.060 | 0.010 | 0.010 | 0.027 | 0.029 | 0.002 | 0.002 | 0.001 | 0.001 | 0.008 | 0.008 | 0.104 | 0.111 |
| Mohammadpur | 0.054 | 0.058 | 0.007 | 0.007 | 0.080 | 0.085 | 0.004 | 0.005 | 0.002 | 0.002 | 0.006 | 0.006 | 0.153 | 0.163 |
| Rayer Bazar | 0.045 | 0.048 | 0.011 | 0.011 | 0.000 | 0.000 | 0.003 | 0.003 | 0.002 | 0.002 | 0.006 | 0.006 | 0.067 | 0.071 |
| Total | 0.326 | 0.346 | 0.038 | 0.039 | 0.107 | 0.114 | 0.015 | 0.016 | 0.007 | 0.008 | 0.033 | 0.036 | **0.527** | **0.561** |

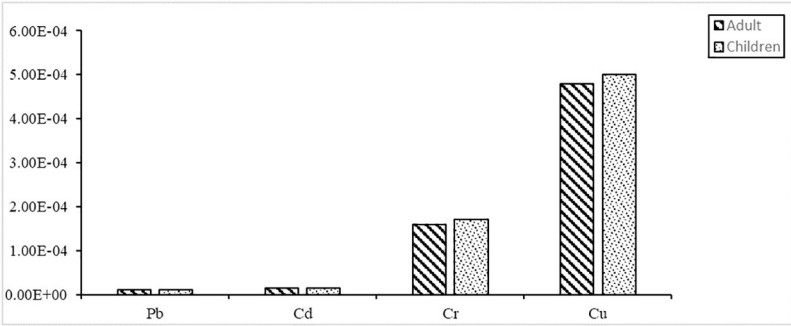

**Fig 3. Target carcinogenic risk (TCR) of Pb, Cd, Cr, and Cu in children and adults from broiler chicken meat consumption.**

and 4.8E-04 for adults 1.0E-05, 1.5E-05, 1.7E-04 and 5.0E-04 for children, respectively. Generally, TCR values less than 1.0E-06 are deemed inconsequential, those more than 1.0E-04 are unsatisfactory, and those between 1.0E-06 and 1.0E-04 are considered acceptable [20, 23]. The carcinogenic risk of Pb, Cd, Cr, and Cu from chicken meat intake was insignificant to tolerable in this study. Chronic Pb exposure from chicken intake that has been enriched with Pb, on the other hand, may put people at risk for cancer in the long term [23].

Our data also imply that children in Bangladesh are more vulnerable than adults to hazardous or non-essential element exposure through regular food intake. Tables 5 and 6 showed that the TCR values for children were, to some extent, higher than that of adults. As a result, the possible health risk to consumers from elemental exposure through eating chicken meals should not be overlooked. This research does not include additional forms of metal exposure, such as eating other meals (such as grains, vegetables, and fish) and inhaling dust. As a result, it is proposed that regular monitoring of harmful and necessary constituents in every food product be carried out to determine whether or not any possible health danger to consumers exists.

## Conclusion

The concentrations of Pb, Cd, Cr, Fe, Cu, and Zn were evaluated on broiler chicken taken from one of the most populated cities in the world, Dhaka, Bangladesh, and the possible risk to human health was assessed using the EDI, THQ, TTHQ, and TCR techniques. All the metal concentrations in the edible body parts of the chicken were within the maximum allowable concentration (MAC) limit, except for Pb and Cu. Pb concentrations exceeded the MAC limit (0.1 mg/kg for meat and 0.5 mg/kg for offal) stipulated by FAO/WHO. The mean concentration of Pb (3.45 mg/kg) in the brain samples was six times higher than the MAC. Cu concentration in the liver, gizzard, heart, kidney and brain exceeded the MAC limit (0.4 mg/kg) specified by FAO/WHO for Cu. Heavy metal EDIs, on the other hand, were significantly lower than the maximum tolerable daily intake (MTDI) limit. The estimated THQ and TTHQ values (0.527 and 0.561) were measured as less than one, suggesting that eating chicken meat poses no non-carcinogenic risk to its consumers. Target carcinogenic risks (TCRs) for Pb, Cd, Cr, and Cu were all within acceptable ranges. The predicted risk assessment for human health showed that the meat of chicken and other edible chicken body parts might be a potential source of particular protein for consumers in terms of contamination with heavy metals. The specific origin of these metals has to be determined by further research. To detect heavy metals in edible chicken body parts in Bangladesh, regular monitoring of feed materials and feeds should serve as the basis for quality control.

## Supporting information

**S1 File.**
(ZIP)

## Acknowledgments

The authors would like to express their most significant appreciation to Jahangirnagar University for initiating the research. Agrochemical and Environmental Research Unit, Institute of Food and Radiation Biology, Atomic Energy Research Establishment, Bangladesh Atomic Energy Commission is highly acknowledged for permitting the laboratory facilities and their overall support throughout the current investigation. We appreciate the opportunity; the editors and reviewers give us to improve the manuscript.

## Author Contributions

**Conceptualization:** Meherun Nesha, Muhammed Alamgir Zaman Chowdhury.

**Data curation:** Easmin Hossain.

**Formal analysis:** Easmin Hossain, Meherun Nesha.

**Investigation:** Easmin Hossain, Meherun Nesha.

**Methodology:** Meherun Nesha, Muhammed Alamgir Zaman Chowdhury.

**Supervision:** Muhammed Alamgir Zaman Chowdhury, Syed Hafizur Rahman.

**Validation:** Meherun Nesha.

**Writing – original draft:** Easmin Hossain.

**Writing – review & editing:** Muhammed Alamgir Zaman Chowdhury, Syed Hafizur Rahman.

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
