## [Decision Letter · Decision Letter 0]

2 Oct 2022

PONE-D-22-23969Quantitative analysis of heavy metals and trace elements in the edible body parts of the chicken and risk assessment of human health

PLOS ONE

Dear Dr. Syed Hafizur Rahman,

Thank you for submitting your manuscript to PLOS ONE. After careful consideration, we feel that it has merit but does not fully meet PLOS ONE’s publication criteria as it currently stands. Therefore, we invite you to submit a revised version of the manuscript that addresses the points raised during the review process.

Major flaws were identified during the reviewing process regarding 2 criteria for publication in PLOS ONE.

Criteria 3: Experiments, statistics, and other analyses are performed to a high technical standard and are described in sufficient detail.

Criteria 4. Conclusions are presented in an appropriate fashion and are supported by the data.

In addition, several weaknesses in the manuscript writing were highlighted. They should all be addressed in order to get a manuscript suitable for publication in PLOS ONE.

We look forward to receiving your revised manuscript.

Kind regards,

Mathilde Body-Malapel

Academic Editor

PLOS ONE

 “No manuscripts currently in press or under consideration elsewhere.

The author(s) received no specific funding for this work. “

“NO authors have competing interests”

4. Please update your submission to use the PLOS LaTeX template. The template and more information on our requirements for LaTeX submissions can be found at http://journals.plos.org/plosone/s/latex.

Reviewers' comments:

Reviewer's Responses to Questions

**Comments to the Author**

1. Is the manuscript technically sound, and do the data support the conclusions?

Reviewer #1: Yes

Reviewer #2: No

2. Has the statistical analysis been performed appropriately and rigorously? 

Reviewer #1: No

Reviewer #2: I Don't Know

3. Have the authors made all data underlying the findings in their manuscript fully available?

Reviewer #1: Yes

Reviewer #2: Yes

4. Is the manuscript presented in an intelligible fashion and written in standard English?

Reviewer #1: Yes

Reviewer #2: No

5. Review Comments to the Author

Reviewer #1: 1.The title is too long

2. conclusion is not true according result of this study (this study demonstrated that

consumers are chronically exposed to elemental contamination with cancer-causing and noncarcinogenic

effects.) because (The calculated THQ

and total target hazard quotient (TTHQ) values were measured less than one, suggesting that the

consumption of chicken meat has no carcinogenic danger to its consumers) please re write this conclusion.

3. for improve the introduction section, please refer to the following related studies:

;https://doi.org/10.1080/03067319.2020.1743835.

https://doi.org/10.1016/j.envres.2021.112002.

https://doi.org/10.1080/10807039.2018.1460798.

https://doi.org/10.3390/ijerph16214261

4. demonstrate novelty in the end paragraph part of introduction .

5.Where are these analyzes used and where are their results reported? A one-way analysis

of variance (ANOVA) test was used to get levels of heavy metal variations in the tested chicken

edible body parts that were statistically significant. The t-test was used to determine the statistically

significant differences between the heavy metals.

What’s the sampling strategy?

6. use this reference in method section :

Health risk assessment of heavy metals in cosmetic products sold in Iran: the Monte Carlo simulation.

Health risk assessment due to fluoride exposure from groundwater in rural areas of Agra, India: Monte Carlo simulation .

7.The conclusion should be rewritten. It is essential to rewrite the conclusion to summary primary results, primary viewpoint, shortcomings, academic contribution, and some suggestion.

Reviewer #2: The paper, Ref. No. PONE-D-22-23969, is basically very week in terms of several technical aspects. The authors have carried out a handsome amount of work but their presentation is very weak. There are several shortcomings in the paper as mentioned hereunder;

1. Some non-technical words such as dangerous heavy metals may be replaced by toxic heavy metals, cancer-causing by carcinogenic and so on.

2. Heavy metals have a specific weight greater than 5 g/cm3 and are naturally present in the Earth’s crust. Ref. is missing here insert (https://doi.org/10.1080/02772248.2017.1413652)

3. Ref. 3 for the sentence “Heavy metals linger in the environment for a long time because they are difficult to break down” is not appropriate, the correct reference could be https://doi.org/10.1155/2019/6730305

4. Check symbols of the metals being studied, throughout the text. Wrote their full name if you want, at their first mention, in the following text only symbols are enough.

5. Citation style in the text is not appropriate, please check throughout the entire text. For instance, “e.g., [2], [7], [9], [14], [15], [19], [26], [27], [29], [31], [32], [37]” references for one statement are too much. Although it is less than [59] Iwegbue et al. (2008) (0.013.22 mg.kg-1), the Pb concentration in chicken gizzards from the Rayer Bazar region was greater than those in chicken gizzards from any other location, check and rephrase; with the study of [37]; according to [38]; in [39]; in [36].

“obtained by [59] Iwegbue et al., (2008)(6.12-33.21 mg/kg), [55] Alturiqi & Albedair, (2012)(27.93-36.93 mg/kg), and [61] Rahman et al., (2014) (26.67±0.25-28.67±0.34 mg/kg).”

6. At various places the word “percent” must be replaced by %, check all other units according standard international protocol. per Sq. Km. is better to be replaced by km-2.

7. “Whatman filter paper (number 42) after the samples” check specification of the filter paper and represent in line with other reports.

8. Table 1, contains some columns which is mere repetition of the identical statements, omit these columns, column 5 and 7 needs to be deleted.

9. Table 2 is very short better to merge it in the text.

10. Some sentences are very short, such as “Lead is known to be a carcinogen”

11. Mostly the table is empty, what is need of such tables? Remove or modify only for valued data, Its removal is highly recommended

12. Reference position of Table 5 is appropriate not in Table 4.

13. Punctuations at several places are inappropriate such as “results of [47]. [14], but” and several other places.

14. Some numerical data is needed to be included in the conclusion.

15. Check references for uniform format and remove some irrelevant references like Ref. [3] does not fit to the statement.

6. PLOS authors have the option to publish the peer review history of their article (what does this mean?). If published, this will include your full peer review and any attached files.

Reviewer #1: No

Reviewer #2: **Yes: **Ezzat Khan

---

## [Author Response · Author response to Decision Letter 0]

22 Nov 2022

Comments from Reviewer #1

 • Comment 1: The title is too long

Response: Thank you for highlighting this. We agree with this comment; therefore, we have changed the title to ‘Human health risk assessment of edible body parts of chicken through heavy metals and trace elements quantitative analysis.’ This addition can be found on page 01 and lines 1–3 in the ‘Revised Manuscript with Track Changes.docx’.

• Comment 2: conclusion is not true according result of this study (this study demonstrated that consumers are chronically exposed to elemental contamination with cancer-causing and noncarcinogenic effects.) because (The calculated THQ and total target hazard quotient (TTHQ) values were measured less than one, suggesting that the consumption of chicken meat has no carcinogenic danger to its consumers) please re write this conclusion.

Response: Thank you for pointing this out. We agree with this comment and have rewritten the conclusion section. This change can be found on pages 25-26, lines 455–472 in the ‘Revised Manuscript with Track Changes.docx’.

• Comment 3: for improve the introduction section, please refer to the following related studies: https://doi.org/10.1080/03067319.2020.1743835. https://doi.org/10.1016/j.envres.2021.112002.

https://doi.org/10.1080/10807039.2018.1460798. https://doi.org/10.3390/ijerph16214261

Response: We agree with this comment and have incorporated suggested references in the introduction part. These changes can be found on page 3, lines 47-63, and page 10, lines 193 in the ‘Revised Manuscript with Track Changes.docx’.

• Comment 4: demonstrate novelty in the end paragraph part of introduction.

Response: Thank you for pointing this out. We agree with this comment and have rewritten the end paragraph part of the introduction. This change can be found on page 05, lines 87-95 in the ‘Revised Manuscript with Track Changes.docx’. 

• Comment 5: Where are these analyzes used and where are their results reported? A one-way analysis of variance (ANOVA) test was used to get levels of heavy metal variations in the tested chicken edible body parts that were statistically significant. The t-test was used to determine the statistically significant differences between the heavy metals. What’s the sampling strategy?

Response: We agree with this comment and have deleted two confusing lines from the manuscript. A multistage cluster sampling strategy was used in the research. 

• Comment 6: use this reference in method section: Health risk assessment of heavy metals in cosmetic products sold in Iran: the Monte Carlo simulation. Health risk assessment due to fluoride exposure from groundwater in rural areas of Agra, India: Monte Carlo simulation.

Response: We agree with this comment and have incorporated the reference in the method section. This change can be found on page 11, line 224 in the ‘Revised Manuscript with Track Changes.docx’.

• Comment 7: The conclusion should be rewritten. It is essential to rewrite the conclusion to summary primary results, primary viewpoint, shortcomings, academic contribution, and some suggestion.

Response: Thank you for pointing this out. We agree with this comment and have rewritten the conclusion section. This change can be found on pages 25-26, lines 455–472 in the ‘Revised Manuscript with Track Changes.docx’.

Comments from Reviewer #2

 • Comment 1: Some non-technical words such as dangerous heavy metals may be replaced by toxic heavy metals, cancer-causing by carcinogenic, and so on.

Response: Thank you for highlighting this. We agree with this comment; therefore, the word ‘dangerous heavy metals’ have been replaced by toxic heavy metals, and ‘cancer-causing’ has been replaced by carcinogenic. These changes have been made throughout the manuscript.

• Comment 2: Heavy metals have a specific weight greater than 5 g/cm3 and are naturally present in the Earth’s crust. Ref. is missing here insert (https://doi.org/10.1080/02772248.2017.1413652)

Response: We agree with this. Accordingly, we have included the reference. This change can be found on page 03, line 47 in the ‘Revised Manuscript with Track Changes.docx’.

• Comment 3: Ref. 3 for the sentence “Heavy metals linger in the environment for a long time because they are difficult to break down” is not appropriate, the correct reference could be https://doi.org/10.1155/2019/6730305

Response: Thank you for pointing this out. We agree with this comment; therefore, the correct reference is used in the revised manuscript. This change can be found on page number 03 and lines 53-54 in the ‘Revised Manuscript with Track Changes.docx’. 

• Comment 4: Check symbols of the metals being studied, throughout the text. Wrote their full name if you want, at their first mention, in the following text only symbols are enough.

Response: We agree. Thank you for highlighting this. Accordingly, the symbols of the metals have been checked and rewritten in the revised manuscript. 

• Comment 5: Citation style in the text is not appropriate, please check throughout the entire text. For instance, “e.g., [2], [7], [9], [14], [15], [19], [26], [27], [29], [31], [32], [37]” references for one statement are too much. Although it is less than [59] Iwegbue et al. (2008) (0.013.22 mg.kg-1), the Pb concentration in chicken gizzards from the Rayer Bazar region was greater than those in chicken gizzards from any other location, check and rephrase; with the study of [37]; according to [38]; in [39]; in [36]. “obtained by [59] Iwegbue et al., (2008)(6.12-33.21 mg/kg), [55] Alturiqi & Albedair, (2012)(27.93-36.93 mg/kg), and [61] Rahman et al., (2014) (26.67±0.25-28.67±0.34 mg/kg).”

Response: Thank you for highlighting this. We agree with this comment; therefore, the citation style in the text has also been changed in the revised manuscript. This change can be found on page number 06, line 126 in the ‘Revised Manuscript with Track Changes.docx’.

• Comment 6: At various places the word “percent” must be replaced by %, check all other units according standard international protocol. per Sq. Km. is better to be replaced by km-2.

Response: We agree with this and have incorporated your suggestion; “percent” has been replaced by % and per Sq. Km. has been replaced by km-2.

• Comment 7: “Whatman filter paper (number 42) after the samples” check specification of the filter paper and represent in line with other reports.

Response: Thank you so much for your comment. We have checked, and the number is 42.

• Comment 8: Table 1, contains some columns which is mere repetition of the identical statements, omit these columns, column 5 and 7 needs to be deleted.

Response: Thank you for highlighting this. We agree with this comment. Accordingly, we have fixed the table. This change can be found on page 8 and line 164 in the ‘Revised Manuscript with Track Changes.docx’.

• Comment 9: Table 2 is very short better to merge it in the text.

Response: We agree with you. Thank you for highlighting this. Therefore, we have merged it in the text. This change can be found on page 10 and lines 199-200 in the ‘Revised Manuscript with Track Changes.docx’.

• Comment 10: Some sentences are very short, such as “Lead is known to be a carcinogen”

Response: We agree with this and have rewritten the sentences. This change can be found on page 15 and line 259 in the ‘Revised Manuscript with Track Changes.docx’.

• Comment 11: Mostly the table is empty, what is need of such tables? Remove or modify only for valued data, It”s removal is highly recommended

Response: Thank you for highlighting this. We modified the table for valued data. 

• Comment 12: Reference position of Table 5 is appropriate not in Table 4.

Response: We agree with this and have corrected it accordingly. This change can be found on page 18 and line 332 in the ‘Revised Manuscript with Track Changes.docx’.

• Comment 13: Punctuations at several places are inappropriate such as “results of [47]. [14], but” and several other places.

Response: Thanks for the observation. We agree with this and have checked and corrected the punctuation throughout the manuscript. 

• Comment 14: Some numerical data is needed to be included in the conclusion.

Response: We agree with this and have incorporated some numerical data in the conclusion section. This change can be found on pages 25-26, lines 455–472 in the ‘Revised Manuscript with Track Changes.docx’.

• Comment 15: Check references for uniform format and remove some irrelevant references like Ref. [3] does not fit to the statement.

Response: We agree with the observation. A relevant reference has replaced the reference [3]. This change can be found on page 03 and lines 53–54 in the ‘Revised Manuscript with Track Changes.docx’.

---

## [Decision Letter · Decision Letter 1]

29 Nov 2022

Human health risk assessment of edible body parts of chicken through heavy metals and trace elements quantitative analysis

PONE-D-22-23969R1

Dear Dr. syed Hafizur Rahman,

We’re pleased to inform you that your manuscript has been judged scientifically suitable for publication and will be formally accepted for publication once it meets all outstanding technical requirements.

Kind regards,

Mathilde Body-Malapel

Academic Editor

PLOS ONE

Additional Editor Comments (optional):

Reviewers' comments:

Reviewer's Responses to Questions

**Comments to the Author**

1. If the authors have adequately addressed your comments raised in a previous round of review and you feel that this manuscript is now acceptable for publication, you may indicate that here to bypass the “Comments to the Author” section, enter your conflict of interest statement in the “Confidential to Editor” section, and submit your "Accept" recommendation.

Reviewer #1: All comments have been addressed

Reviewer #2: All comments have been addressed

2. Is the manuscript technically sound, and do the data support the conclusions?

Reviewer #1: Yes

Reviewer #2: Yes

3. Has the statistical analysis been performed appropriately and rigorously? 

Reviewer #1: Yes

Reviewer #2: Yes

4. Have the authors made all data underlying the findings in their manuscript fully available?

Reviewer #1: Yes

Reviewer #2: Yes

5. Is the manuscript presented in an intelligible fashion and written in standard English?

Reviewer #1: Yes

Reviewer #2: Yes

6. Review Comments to the Author

Reviewer #1: In my opinion, the authors have successfully clarified all the weaknesses of the article, and have managed to answer the questions proposed by the reviewers. Overall, I recommend this manuscript for publication.

Reviewer #2: Authors have addressed all comments and observations raised by this reviewer, to his entire satisfaction, still I advise the authors to read the manuscript carefully to eliminate some of the minor grammatical and technical errors. Reference 55, needs to be changed or replaced, include English version of the reference or replace it with another relevant. If authors are convinced and they understand the language of the paper then OK. Otherwise, it gives an impression that they read, summarized and cited the work. I do not have any major concern to decide against the manuscript.

7. PLOS authors have the option to publish the peer review history of their article (what does this mean?). If published, this will include your full peer review and any attached files.

Reviewer #1: No

Reviewer #2: **Yes: **Ezzat Khan

---

## [Editor Report · Acceptance letter]

13 Dec 2022

PONE-D-22-23969R1 

Human health risk assessment of edible body parts of chicken through heavy metals and trace elements quantitative analysis 

Dear Dr. Rahman:

I'm pleased to inform you that your manuscript has been deemed suitable for publication in PLOS ONE. Congratulations! Your manuscript is now with our production department. 

Kind regards, 

on behalf of

Dr. Mathilde Body-Malapel 

Academic Editor

PLOS ONE